# Cluster-Learngene: Inheriting Adaptive Clusters for Vision Transformers

**Qiufeng Wang**[1,2], **Xu Yang**[1,2†], **Fu Feng**[1,2], **Jing Wang**[1,2], and **Xin Geng**[1,2†]

[1]School of Computer Science and Engineering, Southeast University, Nanjing 210096, China
[2]Key Laboratory of New Generation Artificial Intelligence Technology and Its Interdisciplinary Applications, Southeast University, Ministry of Education, China
`{qfwang, xuyang_palm, fufeng, wangjing91, xgeng}@seu.edu.cn`

## Abstract

In recent years, the merging of vast datasets with powerful computational resources has led to the emergence of large pre-trained models in the field of deep learning. However, the common practices often overgeneralize the applicability of these models, overlooking the task-specific resource constraints. To mitigate this issue, we propose **Cluster-Learngene**, which effectively clusters critical internal modules from a large ancestry model and then inherits them to initialize descendant models of elastic scales. Specifically, based on the density characteristics of attention heads, our method adaptively clusters attention heads of each layer and position-wise feed-forward networks (FFNs) in the ancestry model as the learngene. Moreover, we introduce priority weight-sharing and learnable parameter transformations that expand the learngene to initialize descendant models of elastic scales. Through extensive experimentation, we demonstrate that Cluster-Learngene not only is more efficient compared to other initialization methods but also customizes models of elastic scales according to downstream task resources.

## 1 Introduction

The evolution of deep learning has been profoundly influenced by the confluence of expansive data sources and robust computational capabilities. This collaboration has given rise to large pre-trained foundation models [11, 10, 41, 5], particularly those built upon the Transformer [47, 11], such as the Vision Transformers (ViTs) [11]. The pre-trained foundation models, being widely deployed in various devices like smartphones or edge devices, serve as the initialization point [16, 2, 19, 64, 49, 50] for diverse downstream applications. However, this dominant methodology implicitly assumes that a one-size-fits-all approach, *i.e.*, the whole foundation model is universally apt for every application, neglecting the specific resource constraints (*e.g.*, memory, FLOPs, or latency) inherent to certain downstream tasks. Furthermore, not all tasks demand the full power of these extensive foundation models. This naturally raises a pivotal question: *Can we extract and harness the condensed part of these foundation models to achieve a harmonious balance between accuracy and resource efficiency?*

To achieve the goal of efficiently initializing models, [49, 50] introduce the innovative *Learngene* framework inspired by the observation of genes (cf. Fig. 1 (a)). As showcased in Fig. 1 (b), *Learngene* framework is designed in two pivotal stages. In the first stage, the significant knowledge is condensed from a large **ancestry model** into a more compact part termed as **learngene**. In the next stage, this learngene is inherited to initialize the **descendant models of elastic scales**. [*] Previous works [49, 50]

---

[†]Corresponding authors.

[*]The terms "foundation model" and "ancestry model," as well as "downstream model" and "descendant model," are interchangeably utilized unless distinctions are explicitly mentioned.

38th Conference on Neural Information Processing Systems (NeurIPS 2024).

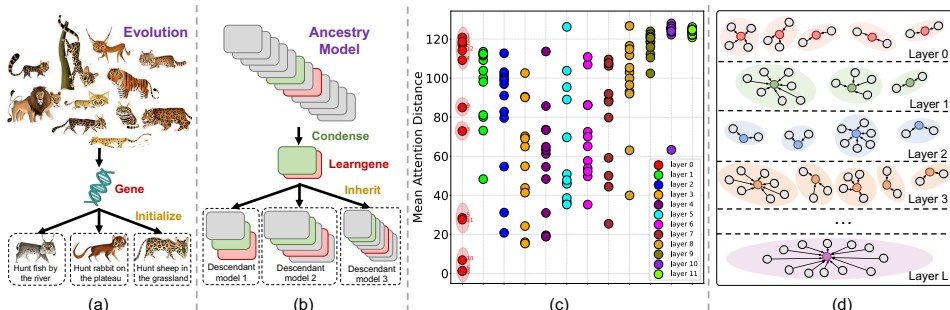

Figure 1: (a) The ancestry of biological organisms condenses evolutionary information into information-dense genes to initialize their diverse descendants [62, 17]. (b) The *Learngene* framework condenses the significant knowledge from an ancestry model into a more compact part termed learngene and then inherited to initialize the descendant models of elastic scales. (c) The density of attention heads across the different layers of the ancestry model, which employs the DeiT-B [46]. (d) An illustration of our idea.

predominantly focus on extracting a few integral layers as the learngene and manually stacking them with the randomly initialized layers.

However, such approaches struggle with inherent limitations: (i) The strategy of extracting certain integral layers overlooks the potential existence of learngene within these layers, leading to the preservation of many redundant weights. (ii) The approach of manually stacking the learngene with randomly initialized layers lacks the adaptability to scale the model, preventing the initialization of downstream models with custom dimensions.

As mentioned earlier, the *Learngene* framework aims to preserve the most generalizable part of the ancestry model while eliminating redundant weights that weaken representational capacity. Recent studies [42, 56] have visualized the mean attention distance of ViTs, offering deeper insights into weight redundancy among attention heads across different layers. As illustrated in Fig. 1 (c), the lower layers focus on both local and global perspectives, leading to a more sparse density of attention heads. Conversely, the higher layers prioritize a global context, resulting in a compact density. A notable observation is the repetitive functionality across many attention heads especially in the higher layers, which inevitably leads to weight redundancy.

Inspired by the above observation, we propose the Cluster-Learngene, an innovative approach that adaptively extracts internal modules in ViTs as the learngene, *e.g.*, attention heads and position-wise feed-forward networks (FFNs). Firstly, to extract the cluster centroids of the attention heads (*i.e.*, **head centroids**) across each layer of the ancestry model, we cluster the attention heads within each layer of the ancestry model based on their density characteristics. As depicted in Fig. 1 (c-d), the attention heads in the first layer exhibit a sparse density, resulting in five clusters, whereas the attention heads in the last layer cluster more compactly, forming a single group. Furthermore, our method includes clustering FFNs by assessing the distance density of head centroids in adjacent layers of the ancestry model. Specifically, when the distance density of head centroids in adjacent layers is similar, we inherit the FFN from the shallower of these adjacent layers (*i.e.*, **FFN centroid**) as the learngene. As illustrated in Fig. 1 (c), the similar densities of attention heads in the 7-th and 8-th layers lead to proximate head centroids, enabling these layers to only require the inheritance from the 7th layer as the FFN centroid. Overall, Cluster-Learngene extracts critical parameters containing significant knowledge, as the extracted part represents attention heads/FFNs with similar semantics.

In the inheriting stage, to achieve the initialization of descendant models with **varying number of attention heads**, we adopt the priority weight-sharing. We start by ranking the head centroids based on the size of their respective clusters, arranging them in descending order of priority. Subsequently, we perform weight-sharing by distributing these head centroids to initialize the attention heads of the descendant models. If the number of attention heads in a specific layer aligns perfectly with the number of centroids, they are evenly shared. However, if they fail to align perfectly, any remaining centroids are shared according to the remainder. Moreover, we apply learnable parameters to transform FFN centroids into multiple FFNs, thus enabling the initialization of descendant models with elastic scales.

Our **contributions** can be summarized as follows: (i) We propose the adaptive clustering to extract the head and FFN centroids as the learngene, ensuring the preservation of significant knowledge within the ancestry model. (ii) To achieve the initialization of descendant models, we introduce priority weight-sharing that favors head centroids within larger clusters and employ learnable parameters to transform the FFN centroids into multiple FFNs. (iii) Comprehensive experimental evaluations across datasets of different scales reveal that Cluster-Learngene not only outperforms traditional initialization strategies but also stands toe-to-toe with more resource-demanding fine-tuning methodologies.

## 2 Related Work

**Model Initialization:**  Over the years, various initialization techniques have been proposed including the popular random initialization, Xavier initialization [13] and the Kaiming initialization [19]. Recently, the use of pre-trained foundation models has gained prominence as an initialization strategy before fine-tuning for specific tasks [11, 10, 41, 57, 37, 5, 61, 22, 15, 53, 33, 32, 7, 18, 39, 54, 29, 25, 26, 51, 28, 44, 55, 59, 40, 58, 36, 31, 21, 60]. However, such an approach necessitates pre-training separate models for each downstream task, which can lead to substantial computational resource consumption. In contrast, Cluster-Learngene presents a unique model initialization method that alleviates the need for multiple pre-training steps.

**Density-based Clustering:** Clustering aims to group similar data points together while separating dissimilar ones. A wide array of approaches has been explored, including partitioning-based clustering [14, 1, 27], hierarchical clustering [35, 8, 65, 52], and density-based clustering [23, 43, 6, 3], and so on. In particular, density-based clustering operates by taking into account the density relationships between data points to form clusters. Inspired by this, our method adopts a similar principle by assessing the density of attention heads to retain essential head centroids that represent significant knowledge.

## 3 Methodology

*Learngene* framework is primarily divided into two phases in Fig. 1 (b): the significant knowledge is condensed from an ancestry model into a more compact part termed as learngene and then inherited to initialize the descendant models of assorted scales. Specifically, in phase 1, our Cluster-Learngene selects mean attention distance as the density metric and uses it to cluster the attention heads of each layer and FFNs in the ancestry model as the **learngene**, because they can effectively represent attention heads/FFNs with similar semantics. The pseudocode for this phase is presented in Algorithm 1. In phase 2, to initialize the descendant models, we employ priority weight-sharing of head centroids for varying number of attention heads as illustrated in Fig.2 and leverage learnable parameters to expand the FFN centroids into multiple FFNs. Next, we briefly introduce some preliminaries related to ViTs.

### 3.1 Preliminary

In the ViT architecture, an input image is first divided into $N$ non-overlapping patches, and each patch is linearly embedded into a flat vector of size $D$. The ViT encoder consists of alternating layers of multi-head self-attention (MSA) and FFN blocks. Let $H$ denote the total number of heads in each layer. For the $h^{th}$ head, the query $\mathbf{Q}_h \in \mathbb{R}^{N \times d_k}$, key $\mathbf{K}_h \in \mathbb{R}^{N \times d_k}$, and value $\mathbf{V}_h \in \mathbb{R}^{N \times d_v}$ are linearly generated through learned weight matrices $\mathbf{W}_h^Q \in \mathbb{R}^{D \times d_k}$, $\mathbf{W}_h^K \in \mathbb{R}^{D \times d_k}$, and $\mathbf{W}_h^V \in \mathbb{R}^{D \times d_v}$, where $d_k$ and $d_v$ are the dimensions of the key and value vectors, respectively. The SA mechanism of the $i$-th head can be represented as:

$$\mathbf{A}^h = \text{Attention}(\mathbf{Q}_h, \mathbf{K}_h, \mathbf{V}_h) = \text{softmax}\left(\frac{\mathbf{Q}_h \mathbf{K}_h^\top}{\sqrt{d_k}}\right) \mathbf{V}_h. \tag{1}$$

MSA allows the model to jointly attend to information at different positions from different representational subspaces at different positions:

$$\text{MultiHead}(\mathbf{Q}, \mathbf{K}, \mathbf{V}) = \text{Concat}(\mathbf{A}^1, \ldots, \mathbf{A}^H)\mathbf{W}^O, \tag{2}$$

where $\mathbf{W}^O \in \mathbb{R}^{H d_v \times D}$ is a learned weight matrix. Besides, the FFN can be formulated as:

$$\text{FFN}(\mathbf{x}) = \text{ReLU}(\mathbf{x}\mathbf{W}_1 + \mathbf{b}_1)\mathbf{W}_2 + \mathbf{b}_2, \tag{3}$$

---

**Algorithm 1:** Adaptively Cluster for MSA

---

1  **Input:** Number of layers as $L$, set of attention heads in the $l^{th}$ layer as $S_l$, radius as $Eps$, density threshold as $MinHds$, and distance function as $Dist$.

2  **Output:** The centroids of attention head in all clusters.

3  Initialize all attention heads as unvisited and an empty list for clusters

4  **for** $l = 1, \dots, L$ **do**

5     **foreach** *attention head $a$ in $S_l$* **do**

6         **if** *$a$ is not visited* **then**

7             Mark $a$ as visited, $NeighborHds \leftarrow$ all attention heads within $Eps$ distance of $a$

8         **end**

9         **if** *number of $NeighborHds \geq MinHds$* **then**

10            $C \leftarrow$ new cluster, Add $a$ to cluster $C$          // Start a new cluster

11            **foreach** *attention head $b$ in $NeighborHds$* **do**

                 // Expand neighborhood

12                **if** *$b$ is not visited* **then**

13                    Mark $b$ as visited

14                    $NeighborHds' \leftarrow$ all attention heads within $Eps$ distance of $b$

15                **end**

16                **if** *number of $NeighborHds' \geq MinHds$* **then**

17                    $NeighborHds = NeighborHds \cup NeighborHds'$

18                **end**

19                **if** *$b$ is not yet a member of any cluster* **then** Add $b$ to cluster $C$

20            **end**

21            Add $C$ to the list of clusters          // Consolidate clusters

22         **end**

23     **end**

24  **end**

---

where $\mathbf{x} \in \mathbb{R}^{N \times D}$ is the input, $\mathbf{W}_1 \in \mathbb{R}^{D \times d_{ff}}$ and $\mathbf{W}_2 \in \mathbb{R}^{d_{ff} \times D}$ are the weight matrices, and $\mathbf{b}_1 \in \mathbb{R}^{d_{ff}}$ and $\mathbf{b}_2 \in \mathbb{R}^D$ are the bias vectors. $d_{ff}$ is the dimension of the intermediate layer.

### 3.2   Adaptively Learngene Clustering

#### 3.2.1   Density metric on attention heads

Given a pre-trained ancestry model with $L$ layers and $H_a$ attention heads per layer, let the attention weights for the $h^{th}$ head in the $l^{th}$ layer be denoted by the matrix $\mathbf{A}^{(l,h)} \in \mathbb{R}^{N \times N}$. The element $A_{i,j}^{(l,h)}$ represents the attention weight from position $i$ to position $j$. The distance matrix $\mathrm{T} \in \mathbb{R}^{N \times N}$ is defined with the Euclidean distance between any two positions $i$ and $j$ in the sequence, given by $T_{i,j} = \sqrt{(x_i - x_j)^2 + (y_i - y_j)^2}$. The mean attention distance for the $h^{th}$ head in the $l^{th}$ layer, encapsulating the weighted distance for each position $i$ across the sequence, is given by:

$$\tilde{d}^{(l,h)} = \frac{1}{N} \sum_{i=1}^{N} \sum_{j=1}^{N} A_{i,j}^{(l,h)} \times T_{i,j}. \tag{4}$$

To deduce this metric for every head across all layers, iterate the above computation for every $l \in \{1, \dots, L\}$ and $h \in \{1, \dots, H_a\}$. As depicted in Fig. 1 and Appendix A.1, while the lower layers simultaneously attend to both local and global features, leading to a more dispersed distribution of attention heads, the higher layers predominantly focus on global aspects, causing a tighter concentration of attention heads. As a result, there is a significant overlap in the semantic representations among many attention heads, especially in the higher layers, leading to weight redundancy.

#### 3.2.2   Cluster for MSA

Motivated by the empirical observations, we extract cluster centroids [43, 6, 3] of attention heads in ViTs as the learngene inherited into the descendant models, thus aggregating similar semantics

into the head centroids. To realize this, we select $\tilde{d}$ as a density metric for adaptively clustering the attention heads of the ancestry model at each layer, without setting the number of clusters in advance. This realization prompts the formulation of the definitions and lemmas, which scaffold our adaptive clustering approach.

**Definition 1** *(Eps-neighborhood of an attention head). The Eps-neighborhood of an attention head a, denoted as $N_{Eps}(a)$, is defined as: $N_{Eps}(a) = \{b \in S \mid Dist(a,b) \leq Eps\}$, where $Dist(a,b) = \left| \tilde{d}^{(a)} - \tilde{d}^{(b)} \right|$ denotes the difference in $\tilde{d}$ values between attention heads a and b.*

Our approach could require for each head in a cluster that there are at least a **Min**imum number of **He**ads ($MinHds$) in an Eps-neighborhood of that head.

**Definition 2** *(density-reachable). Transitioning from the neighborhood concept, an attention head a is considered density-reachable from another head b with respect to $Eps$ and $MinHds$ if there is a sequence of heads $a_1, \ldots, a_n$ such that $a_1 = b, a_n = a$, and each head in this sequence lies within the Eps-neighborhood of its preceding head.*

**Definition 3** *(density-connected). Broadening our purview, attention heads a and b are labeled density-connected with respect to $Eps$ and $MinHds$ if there exists an intermediary head o from which both a and b are density-reachable.*

Considering all attention heads in layer $l$ as $S_l$, a cluster $C$ based on $Eps$ and $MinHds$ is identified as a non-empty subset of $S_l$ that satisfies the conditions: (i) **Maximality:** For any heads $a$ and $b$ in the sequence, if $a$ resides within $C$ and $b$ is *density-reachable* from $a$ dictated by $Eps$ and $MinHds$, then $b$ seamlessly becomes part of $C$. (ii) **Connectivity**: Within $C$, each pairing $a, b$ maintains a *density-connection*, anchored by $Eps$ and $MinHds$.

Therefore, upon satisfying these two conditions, we cluster all attention heads of each layer into different lists of clusters. The pseudo-code for this process is summarized in Algorithm 1. For each cluster $C$ in the list of clusters, we select the attention head $C_{lg}$ closest to the cluster centroid as the learngene. This is because it effectively represents the functionality of all attention heads in the cluster. Formally, the formula can be expressed as follows:

$$C_{lg} = \arg \min_{c \in C} \left| \tilde{d}^{(c)} - \bar{d}^{(C)} \right| \tag{5}$$

where $\bar{d}^{(C)}$ is computed as the average $\tilde{d}$ across all attention heads within this cluster $C$. The lemma presented below is pivotal in substantiating the correctness of our clustering algorithm.

**Lemma 1** *Presuming an attention head a belongs to $S_l$ and satisfies the condition $|N_{Eps}(a)| \geq MinHds$. Then, the set $O = \{o \mid o \in S_l \text{ and } o \text{ is density-reachable from } a \text{ with respect to } Eps \text{ and } MinHds\}$ collectively shapes a cluster.*

### 3.2.3 Cluster for FFN

Based on the distances between attention heads discussed earlier, we can further cluster FFNs as the learngene. For adjacent layers sharing the same number $N_C$ of head centroids, a key criterion is established: if the average distance across all head centroids between these layers is less than the threshold $\varepsilon$, it suggests redundant similarity in their representational capabilities. This criterion is formulated as:

$$\frac{1}{N_C} \sum_{k=1}^{N_C} Dist\left(C_{l,k}, C_{l+1,k}\right) < \varepsilon. \tag{6}$$

Under this condition, we enhance model efficiency by preserving only the FFN from the shallower of these adjacent layers as the FFN centroid, thereby maintaining essential functionality while eliminating redundancy. Moreover, we also select those FFNs that do not fit this established criterion as the learngene because they exhibit representational capacities independent of adjacent layers.

### 3.3 Learngene Inheriting

#### 3.3.1 Expanding head centroids with priority weight-sharing

Since the head centroids have been extracted as the learngene, we further expand them for initializing the descendant models with varying number of attention heads. For an ancestry model with $L$

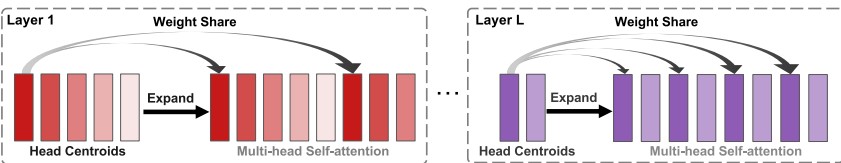

Figure 2: Illustration of priority weight-sharing. The darker the color, the larger the cluster size associated with the head centroid.

layers, the $l^{th}$ layer has $c_l$ head centroids of weight $\mathbf{A}^{(l,1)}, \ldots, \mathbf{A}^{(l,c_l)}$. Importantly, the head centroids at each layer are sorted in descending order based on the size of their respective cluster, *i.e.*, centroids representing more attention heads in the ancestry model are ranked higher. These head centroids condense significant knowledge and ensure the initialization of descendant models without performance degradation. Assume the descendant model has $H_d$ attention heads for each layer. To achieve the desired expansion of heads to initialize the descendant models, we adopt the priority weight-sharing and Fig. 2 illustrates two scenarios:

- When $H_d$ is divisible by $c_l$: The weights of head centroids are shared $\frac{H_d}{c_l}$ times in sequence. For instance, centroids of weights $\mathbf{A}^{(L,1)}$ and $\mathbf{A}^{(L,2)}$ each share their weights across four attention heads, which are then directly assigned to eight attention heads of the descendant model in layer $L$.

- When $H_d$ is not divisible by $c_l$: The weights of the head centroids are sequentially shared $\left\lfloor \frac{H_d}{c_l} \right\rfloor$ times, followed by appending $\mathbf{A}^{(l,1)}, \ldots, \mathbf{A}^{(l,H_d \mod c_l)}$ at the end. As an illustration, we share the centroids of weights $\mathbf{A}^{(1,1)}, \ldots, \mathbf{A}^{(1,5)}$ once and then append $\mathbf{A}^{(1,1)}, \ldots, \mathbf{A}^{(1,3)}$, thus initializing eight attention heads of the descendant model in the first layer.

For the attention heads in the descendant models, we introduce the hyperparameter $\omega = \frac{H_a}{H_d}$ to denote the factor by which the number of attention heads is reduced compared to the ancestry model. In addition to uniformly setting the number of attention heads for each layer with the hyperparameter $\omega$, we also explore two other possibilities in Appendix A.7: incrementing and decrementing the count of attention heads with layer depth. According to the adjustments in the number of attention heads, the weights $\mathbf{W}^O$ of the projection layer are also proportionally pruned and then inherited by the descendant models. [†]

### 3.3.2 Expanding FFN

As outlined in Section 3.2, we retain the shallowest FFN determined by distance-based clustering. For this FFN centroid, we introduce learnable parameters to transform it into multiple FFNs, facilitating the initialization of descendant models at elastic scales. This process is summarized in the formula:

$$\text{FFN}_t(\mathbf{x}) = (\text{ReLU}((\mathbf{x}(\mathbf{W}_1 + \mathbf{b}_1)\widehat{\mathbf{W}}_t)\mathbf{W}_2 + \mathbf{b}_2)\widehat{\mathbf{W}}_t, \tag{7}$$

Here, $\widehat{\mathbf{W}}_t$ denotes the newly introduced learnable parameters to expand the $t^{th}$ FFN. These parameters are designed for minimal learning overhead, yet they are highly effective in quickly adapting descendant models to downstream tasks.

## 4 Experiments

### 4.1 Experimental Setting

**Datasets.** To condense the learngene, we employ the ImageNet-1K, a collection of 1.2 million training images and 50,000 validation images distributed across 1,000 classes as part of the ILSVRC2012 competition [9]. After initializing the descendant models with the learngene, we proceed to fine-tune these models on diverse downstream tasks. These tasks include iNaturalist-2019 [45], Food101 [4], Oxford Flowers [38], Stanford Cars [12], CIFAR-10 [24], CIFAR-100 [24], CUB-200-2011 [48]. For detailed dataset descriptions, see Appendix A.2.

---

[†]Please see Appendix A.3 for more details.

**Training settings.** During the learngene clustering, We set $Eps, \varepsilon = 10, MinHds = 1$ [‡], ensuring that each attention head is included in a unique cluster. In the learngene inheriting phase, we train the descendant models on downstream tasks for 500 epochs, including a 10-epoch warm-up period, except for iNaturalist-2019, where we train for 100 epochs with a 5-epoch warm-up. The initial learning rate is set to $5 \times 10^{-4}$ for most tasks, except for Stanford Cars where it is $5 \times 10^{-3}$, and a weight decay of 0.05.

**Architectures.** Both the ancestry model and descendant models are variants derived from DeiT [46]. In terms of width, there are three types of DeiT: **Tiny**, **Small**, and **Base**. Furthermore, as detailed in Section 4.2.4, we implement experiments on Swin Transformer [30] to demonstrate the applicability of our method across various backbones. The learnable parameters in Eqns. (7) are implemented through a nonlinear mapping such as a neural network with the rectified linear units (ReLU).

## 4.2 Main Results of Model Initialization

In this section, we validate the capabilities of Cluster-Learngene in efficiently initializing models and measure model performance with Top-1 accuracy.

### 4.2.1 Initializing Descendant Models of Elastic Scales

As illustrated in Fig. 3, for the Tiny-scale descendant models with only 32 attention heads, Cluster-Learngene outperforms Pretraining-Finetuning on ImageNet. Moreover, the performance of Cluster-Learngene improves with an increase in the number of attention heads and FFNs. This shows that Cluster-Learngene can adaptively initialize the descendant models with varying number of attention heads and FFNs, catering to downstream resource constraints. In contrast, Pretraining-Finetuning requires retraining for each model variant, significantly raising storage and training costs, particularly when dealing with models of elastic scales. Therefore, our method resolves the limitations of the one-size-fits-all approach seen in Pretraining-Finetuning.

Moreover, we initialize 22 descendant models on ImageNet-1K, each with different configurations, such as the number of attention heads $H_d$ per layer and the quantity of FFNs $L_d$ in descendant models. As shown in Tab. 1, Cluster-Learngene swiftly initializes models of varying scales and proves competitive in overall performance. For example, the Small-scale descendant model with $H_d = 6$ and $L_d = 12$ illustrates this point. Cluster-Learngene not only outperforms From-Scratch by **9.87%** but also manages to reduce $2\times$ training times.

Table 1: Comparisons of performance on ImageNet-1K between models trained From-Scratch with 100 epochs and those initialized via Cluster-Learngene fine-tuned for 50 epochs.

| Model | $H_d$ | $L_d$ | Params (M) | FLOPs (G) | From-Scratch | Ours |
|---|---|---|---|---|---|---|
| Tiny | 2 | 6 | 1.3 | 0.3 | 50.06 | **52.73** |
| | | 9 | 1.9 | 0.4 | 54.64 | **59.60** |
| | | 12 | 2.5 | 0.5 | 57.99 | **61.84** |
| | 3 | 6 | 3 | 0.6 | 58.16 | **59.58** |
| | | 9 | 4.4 | 0.9 | 60.58 | **65.47** |
| | | 12 | 5.7 | 1.2 | 61.44 | **70.28** |
| Small | 4 | 6 | 5 | 1 | 61.32 | **64.98** |
| | | 9 | 7.3 | 1.4 | 63.17 | **70.90** |
| | | 12 | 9.5 | 1.9 | 64.25 | **73.20** |
| | 6 | 6 | 11.4 | 2.3 | 64.91 | **67.72** |
| | | 9 | 16.8 | 3.4 | 67.02 | **73.06** |
| | | 12 | 22 | 4.6 | 68.56 | **78.43** |
| Base | 12 | 3 | 22.8 | 4.5 | 68.77 | 67.82 |
| | | 4 | 29.9 | 5.9 | 70.32 | **70.41** |
| | | 5 | 36.9 | 7.4 | 72.04 | **72.13** |
| | | 6 | 44 | 8.8 | 73.73 | **73.95** |
| | | 7 | 51.2 | 10.2 | 74.42 | **74.87** |
| | | 8 | 58.2 | 11.7 | 76.14 | **76.90** |
| | | 9 | 65.3 | 13.1 | 76.46 | **77.11** |
| | | 10 | 72.4 | 14.6 | 76.81 | **77.99** |
| | | 11 | 79.5 | 16 | 77.03 | **78.82** |
| | | 12 | 86.6 | 17.5 | 77.22 | **79.50** |

### 4.2.2 Efficiently Initializing Large Models on ImageNet

The results in Fig. 4 reveal that Cluster-Learngene outperforms Pretraining-Finetuning with fewer inherited parameters. For example, in the case of Small-scale descendant models, Cluster-Learngene inherits only 15.1M parameters and attains a performance of 78.43%, while Pretraining-Finetuning with 18.0M parameters achieves less than 75%. These results demonstrate the advanced initialization ability of Cluster-Learngene, as the clustered learngene retains the critical generalizable knowledge.

### 4.2.3 Initialization Results on Different Downstream Tasks

We conduct a comparative analysis of our approach for initializing descendant or downstream models, as follows: (i) Pretraining-Finetuning: This approach pre-trains DeiT on ImageNet and

---

[‡]Please see Appendix A.6 for more analysis.

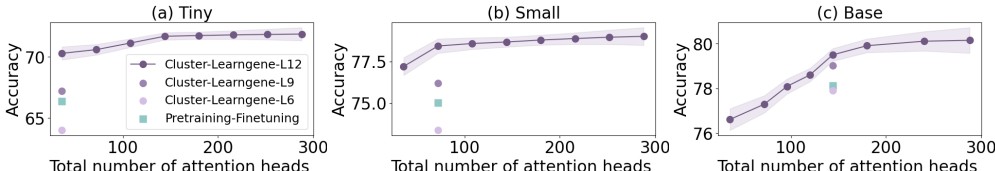

Figure 3: **Initializing descendant models of elastic scales.** "L6/9/12" denote descendant models with 6, 9, and 12 layers, respectively. For a fair comparison, the downstream models in Pretraining-Finetuning inherit parameters from 12 layers of the pre-trained model, with the inherited number of attention heads matching those in Cluster-Learngene. We fine-tune **50 epochs** for all models. In (a), the hyperparameter $\omega$ takes values ranging from 1 to $\frac{1}{8}$ (i.e., the number of attention heads in descendant models is eight times that of the ancestry model). In (b), $\omega$ ranges from 2 to $\frac{1}{4}$. Continuing this pattern, in (c), $\omega$ ranges from a maximum of 4 to a minimum of $\frac{1}{2}$.

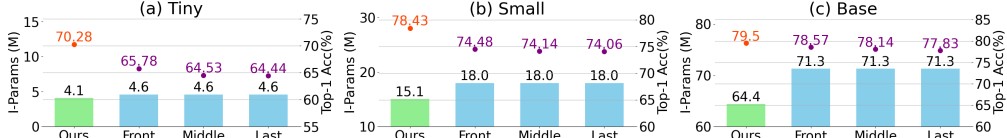

Figure 4: **Efficiently initializing large models on ImageNet.** "Front/middle/last" refer to inheriting parameters from the front, middle, or last 10 layers of a pretrained model to initialize 12-layer descendant models. All approaches are fine-tuned for 50 epochs. "I-Params" means the number of **I**nherited parameters in the downstream/descendant models, measured in MB.

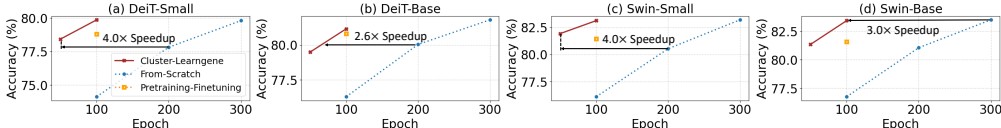

Figure 5: **Faster convergence.** Different points represent results for varying epochs and the hyperparameter $\omega$ is set to 1.0 for our method.

Table 2: **DeiT-Small Results on downstream datasets.** ↑ represents the performance improvement achieved by Cluster-Learngene, when compared to the best method excluding Pretraining-Finetuning. All results are derived from the 6-layer downstream models.

| Method | I-Params | iNat-2019 | Food-101 | Flowers | Cars | CIFAR-10 | CIFAR-100 | CUB-200 |
|---|---|---|---|---|---|---|---|---|
| Pretraining-Finetuning | 10.5 | 68.48 | 87.8 | 91.13 | 86.81 | 97.59 | 84.43 | 78.13 |
| From-Scratch | 0 | 50.79 | 74.64 | 72.91 | 71.63 | 92.49 | 73.32 | 62.75 |
| Heuristic-Learngene | 5.6 | 53.21 | 77.09 | 82.84 | 81.52 | 93.12 | 78.13 | 72.64 |
| Weight-Transformation | 10.5 | 59.83 | 81.79 | 86.37 | 85.01 | 93.67 | 75.98 | 70.28 |
| Auto-Learngene | 10.5 | 59.92 | 80.25 | 87.02 | 84.98 | 93.58 | 79.49 | 73.31 |
| **Cluster-Learngene** | 7.5 | **71.09(↑11.17)** | **89.53(↑7.74)** | **92.31(↑5.29)** | **89.87(↑4.86)** | **97.79(↑4.12)** | **85.38(↑5.89)** | **75.98(↑2.67)** |

subsequently fine-tunes the entire model on downstream tasks. (ii) From-Scratch: We commence with a randomly initialized DeiT model on the downstream datasets. (iii) Heuristic-Learngene [49]: This strategy involves extracting the last six layers from a DeiT model pre-trained on ImageNet and then stacking them with randomly initialized lower layers to construct a new model. (iv) Weight-Transformation [63]: This method employs Weight Transformation to pre-train DeiT on ImageNet, followed by fine-tuning the entire model to adapt it to specific downstream tasks. (v) Auto-Learngene [50]: The first six layers are extracted from the DeiT and then stacked with randomly initialized higher layers to initialize the descendant models.

As illustrated in Tab. 2, our Cluster-Learngene significantly outperforms both From-Scratch and Weight-Transformation. When compared to other Learngene methods, such as Auto-Learngene, Cluster-Learngene exceeds by **11.17%** on the iNaturalist-2019 (iNat-2019). These results highlight the superior capability of Cluster-Learngene in efficiently initializing descendant models. Moreover, on six datasets, the performance of Cluster-Learngene outperforms that of Pretraining-Finetuning, where the entire model is fine-tuned. This phenomenon can be attributed to the more universally significant knowledge within learngene, allowing it to adapt effectively to various downstream tasks.

#### 4.2.4 Faster Convergence

We provide a detailed comparison of training efficiency between our approach and From-Scratch on ImageNet. As shown in Fig. 5 (a), Cluster-Learngene requires only **4.0** × less training overhead compared to From Scratch on Small-scale descendant models. A key advantage of our approach is that descendant models initialized with the learngene achieve faster convergence, owing to a superior initialization point.

#### 4.2.5 Higher Data Efficiency

We further conduct experiments on Base-scale descendant models over different percentages of training data from ImageNet-1K (IN-1K). As shown in Tab. 3, while our method does not outperform the From-Scratch on the entire dataset, its performance exhibits greater stability as the amount of training data decreases. For instance, with only 25% of the training data, Cluster-Learngene outperforms From-Scratch by **7.09%**, while requiring only $\frac{1}{6}$ of the training cost. This higher data efficiency of our method is attributed to the significant knowledge within the learngene, which helps descendant models mitigate overfitting, especially in scenarios with limited data.

Table 3: **Initialization of descendant models with diverse training samples.** The symbol ↑ denotes the performance gap between our approach and the From-Scratch method. Cluster-Learngene initializes the descendant model over 50 training epochs. In contrast, From-Scratch results are achieved after 300 training epochs.

| Training data | From-Scratch | Cluster-Learngene |
|---|---|---|
| 100% IN-1K | 81.80 | 78.43 |
| 50% IN-1K | 74.70 | **76.41**(↑**1.71**) |
| 25% IN-1K | 65.73 | **72.82**(↑**7.09**) |

### 4.3 Analysis and Ablation

In this section, we provide further analysis and ablation of Cluster-Learngene. [§] Unless otherwise specified, we conduct experiments on **CIFAR-100** and use **Small-scale DeiT** as the ancestry model.

#### 4.3.1 Comparison of the Clustering Method

Additionally, we compare the results of *k-means* clustering [20] of attention heads with cluster centroids ($k$) set at 1, 2, and 3. Tab. 4 shows that Cluster-Learngene not only outperforms in clustering efficiency but also adaptively adjusts the count of cluster centroids for each model layer, unlike *k-means* that requires predefined the numbers of cluster centroids.

Table 4: **Comparison of the clustering method.** All results are from the 6-layer downstream models.

| *k-means*, $k = 1$ | *k-means*, $k = 2$ | *k-means*, $k = 3$ | Ours |
|---|---|---|---|
| 80.12 | 81.02 | 81.41 | **85.38** |

#### 4.3.2 Comparison of Priority Weight-sharing

In Table 5, our priority weight-sharing is compared against three alternative weight-sharing methods: following the original sequence of heads, by ascending $\tilde{d}$ of heads, and by descending $\tilde{d}$ of heads. The results validate the effectiveness of priority weight-sharing, as our method accounts for the fact that the larger the cluster a head belongs to, the richer the critical knowledge it represents.

Table 5: **Comparison of priority weight-sharing.** All results are derived from the 6-layer downstream models.

| Origin | Increasing $\tilde{d}$ | Decreasing $\tilde{d}$ | Ours |
|---|---|---|---|
| 83.98 | 84.24 | 84.63 | **85.38** |

#### 4.3.3 Qualitative Visualization

We illustrate attention representations in Fig. 6 to explain which significant knowledge is inherited by learngene. To clarify the visualization, we apply a power exponent of $\gamma = 0.25$. In the first layer of the ancestry model, a head centroid is clustered from heads 1, 2, 4, and 5 to initiate the head 1 in

---

[§]Please see Appendix A.9 and A.10 for more visualization.

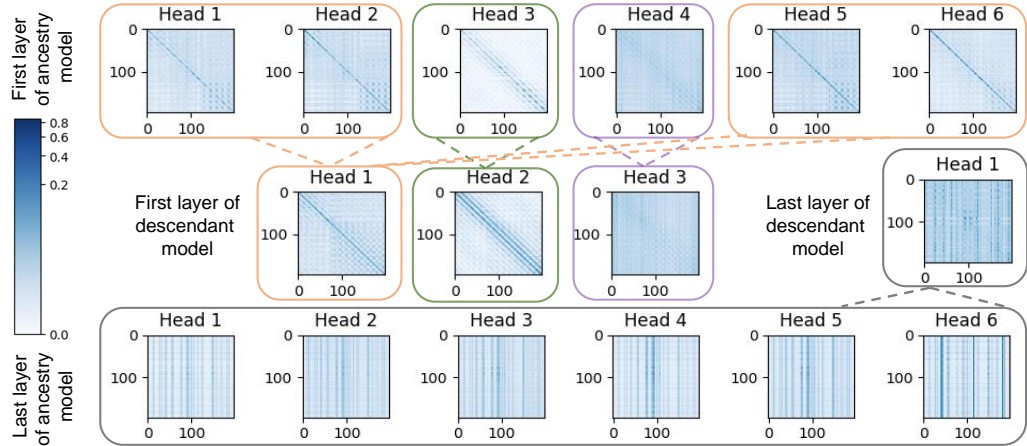

Figure 6: **Visualization of attention representations** $(197 \times 197)$. We perform the following normalization operation on all attention heads $\mathbf{A}$ of the ancestry model and descendant model: $\left(\frac{\mathbf{A}_{i,j}}{255}\right)^{\gamma}$. The descendant model is trained for 50 epochs, and $\omega$ is set to $\frac{1}{4}$.

the descendant model, and so forth. Then, weight-sharing is applied to expand head centroids, *e.g.*, sharing twice to initialize the descendant model.

In the first layer, heads 1, 2, 4, and 5 with similar semantics form the largest cluster, mainly focusing on attention patterns along the main diagonal. Additionally, heads 3 and 4 present distinct semantics, with head 4 reflecting more abstract and high-level features akin to the final layer. Thus, the first learngene layer integrates three principal representation patterns from the ancestry model. In contrast, the representations in the final layer of the ancestry model exhibit significant repetition, leading to the clustering of a single-head centroid for initializing the attention heads of the descendant model.

## 5    Conclusion

We propose Cluster-Learngene to adaptively cluster attention heads, extracting head centroids and FFN centroids as the learngene. Subsequently, we adopt the priority weight-sharing of head centroids for varying number of attention heads and leverage learnable parameters to expand the FFN centroids into multiple FFNs, enabling adaptation to diverse downstream resource constraints. Extensive experiments validate the efficiency and scalability of our initialization method.

## Acknowledgments and Disclosure of Funding

This research was supported by the National Science Foundation of China (62125602, 62076063), the Key Program of Jiangsu Science Foundation (BK20243012), the Fundamental Research Funds for the Central Universities (2242024k30035) and the Big Data Computing Center of Southeast University.

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

| Dataset | # Total | #Training | #Validation | #Testing | #Classes |
|---|---|---|---|---|---|
| Oxford Flowers [38] | 8,189 | 1,020 | 1,020 | 6,149 | 102 |
| CUB-200-2011 [48] | 11,788 | 5,394 | 600 | 5,794 | 200 |
| Stanford Cars [12] | 16,185 | 7,329 | 815 | 8,041 | 196 |
| CIFAR10 [24] | 65,000 | 50,000 | 5,000 | 10,000 | 10 |
| CIFAR100 [24] | 65,000 | 50,000 | 5,000 | 10,000 | 100 |
| Food101 [4] | 101,000 | 75,750 | 25,250 | 0 | 101 |
| iNaturalist-2019 [45] | 268,243 | 265,213 | 3030 | / | 1010 |

Table 6: Characteristics of the downstream datasets

# A  Appendix / supplemental material

## A.1  Mean Attention Distance in DeiT-S and DeiT-Ti

Fig. 7 illustrates the mean attention distance for two other variants of DeiT. In both variants, the lower layers exhibit a dual focus on both local and global aspects, resulting in a relatively sparse distribution of attention heads. Conversely, the higher layers prioritize the global context, leading to a more compact distribution of attention heads. Importantly, many attention heads in these layers exhibit repetitive functionality, contributing to weight redundancy. In the process of FFN clustering, the shallowest layers are preserved as the FFN centroids across different configurations: in DeiT-Tiny, this applies to layers (2,3) and (11,12); in DeiT-Small, to layers (10,11,12); and in DeiT-Base, to layers (7,8) and (10,11,12).

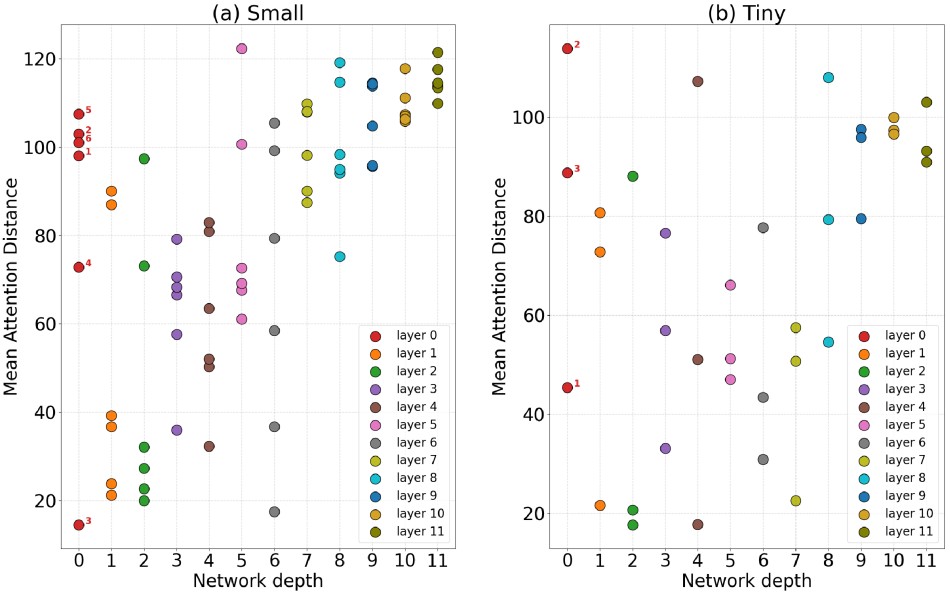

Figure 7: The distribution density of attention heads across the different layers of the ancestry model, which employs the DeiT-S and DeiT-Ti [46].

## A.2  Downstream Datasets

Tab. 6 presents the details of all downstream tasks.

## A.3  Projection Layer

According to the adjustments in the number of attention heads, the weights $\mathbf{W}^O$ of the projection layer are also proportionally pruned or expanded with the hyperparameter $\omega$ and then inherited by

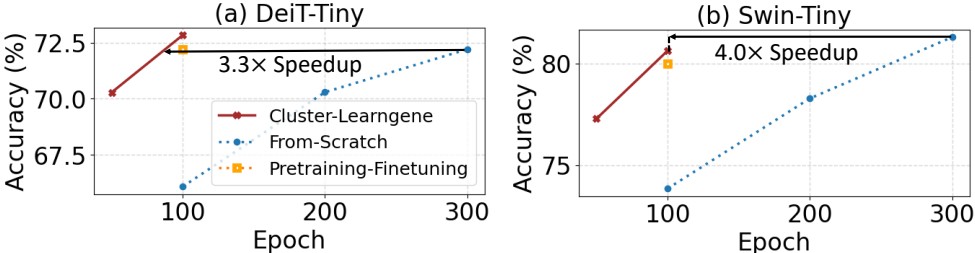

Figure 8: **Faster convergence.** Different points represent results for varying epochs and the hyperparameter $\omega$ is set to 1.0 for our method.

the descendant models. Additionally, we directly inherit the weights of layer normalization, patch embeddings, and position embeddings in the ancestry model, which constitute only a small fraction of all weights.

### A.4 Faster Convergence

We provide a detailed comparison of training efficiency on ImageNet for the Tiny-scale descendant models. As shown in Fig. 8 (b), Cluster-Learngene requires only **4.0** $\times$ less training overhead compared to From Scratch on Small-scale descendant models. A key advantage of our approach is that descendant models initialized with the learngene achieve faster convergence, owing to a superior initialization point.

### A.5 Ablation on the Selection of Head Centroids

We conduct an ablation to assess whether inheriting parameters from the nearest module to the cluster centroid or averaging parameters after clustering heads and FFNs is more effective. Table 7 shows Cluster-Learngene excelling when inheriting from the closest heads/FFNs to the centroid, a logical approach as these modules aptly represent similar semantics within a cluster.

| MSA | FFN | Acc (%) |
|---|---|---|
| average | average | 85.12 |
| argmin | average | 84.29 |
| average | argmin | 84.76 |
| argmin | argmin | **85.38** |

Table 7: **Ablation on the selection of head centroids.** All results are derived from the 6-layer downstream models. we conduct experiments on CIFAR-100 and use DeiT-Small as the ancestry model.

### A.6 Ablation on $\varepsilon$

Table 8 shows the results for different values of $\varepsilon$. $\varepsilon = 1$ implies no FFN clustering, potentially causing negative transfer. $\varepsilon = 100$ means clustering all FFNs in adjacent layers with identical head centroid counts, resulting in an excessive cluster of FFNs and subsequent degradation in the performance of initialized descendant models. Our Cluster-Learngene ($\varepsilon = 10$) strikes a good balance between these issues.

| $\varepsilon = 1$ | $\varepsilon = 10$ | $\varepsilon = 100$ |
|---|---|---|
| 85.27 | **85.38** | 80.45 |

Table 8: **Ablation on $\varepsilon$.** All results are derived from the 6-layer downstream models.

| Model | Decrementing | Incrementing |
|-------|--------------|--------------|
| Tiny | 76.56 | **78.01** |
| Small | 79.47 | **81.29** |
| Base | 80.18 | **81.65** |

Table 9: **Increment or decrement the count of attention heads.** "Decrementing" denotes halving the number of attention heads in the first four layers, reducing them by a quarter in the middle four layers, and maintaining them in the last four layers relative to the ancestry model. Conversely, "Incrementing" represents the opposite pattern.

### A.7  Variation in the Count of Attention Head with Model Depth

Tab. 9 presents two scenarios where the number of attention heads varies across different layers. Across all descendant model configurations, "Incrementing" consistently outperforms "Decrementing" by a margin of 1.45% in terms of accuracy. These findings align with previous research [34, 30], which suggests that setting more attention heads in higher layers can assist these layers in learning more abstract and high-level feature representations.

### A.8  Initializing Descendant Models of Elastic Scales

As illustrated in Fig. 3, Cluster-Learngene incurs a total of 300 + 50 * 10 = 800 epochs. In contrast, pre-trained models require retraining a new model of elastic scale for 300 epochs each, followed by fine-tuning. Considering only the training expense of new models, training 10 such models would require at least 300 * 10 = 3000 epochs. Therefore, our algorithm can reduce the computational cost by at least $3\times$. This efficiency demonstrates how our method overcomes the one-size-fits-all limitation inherent in the Pretraining-Finetuning approach.

### A.9  Limitations

The performance of descendant models within Cluster-Learngene heavily depends on the quality of the ancestry model. If the ancestry model harbors inherent limitations or biases, these issues may propagate and even amplify within the descendant models, potentially compromising their effectiveness. Therefore, ensuring the robustness and fairness of the ancestry model is paramount for maintaining the performance of descendant models in Cluster-Learngene.

