# OpenReview forum: "Cluster-Learngene: Inheriting Adaptive Clusters for Vision Transformers"
_NeurIPS.cc/2024/Conference — NeurIPS 2024 poster_

### Official Review · Reviewer_ScsP · 2024-07-04

**Soundness:** 2
**Presentation:** 2
**Contribution:** 2
**Rating:** 5
**Confidence:** 3

**Summary:**

Cluster-Learngene is an innovative approach for initializing Vision Transformer models. It works by inheriting adaptive clusters from a large pre-trained ancestry model. The key features of this method include:
Adaptive Clustering: Cluster-Learngene adaptively clusters the attention heads and feed-forward networks within each layer of the ancestry model based on their density characteristics. This forms compact and information-dense modules called "learngenes."
Efficient Initialization: The method uses priority weight-sharing and learnable parameter transformations to expand these learngenes. This enables efficient initialization of descendant models with varying scales and resource constraints.
Resource Customization: Cluster-Learngene tailors the model initialization to the downstream task resources, offering a more efficient alternative to the standard pretraining and fine-tuning approach.
Empirical Validation: Extensive experiments on multiple datasets demonstrate that Cluster-Learngene outperforms traditional initialization strategies and is competitive with more resource-intensive fine-tuning methods.
Scalability and Efficiency: The approach achieves faster convergence and higher data efficiency compared to models initialized from scratch, making it effective for scenarios with limited data.
Broad Applicability: The paper shows the effectiveness of Cluster-Learngene across different model scales and various downstream tasks, showcasing its versatility.
In summary, Cluster-Learngene presents a novel and efficient strategy for initializing Vision Transformer models, addressing the challenge of resource constraints in downstream tasks.

**Strengths:**

This paper contributes to the field of deep learning with several key strengths:

Originality: It proposes an innovative method for initializing Vision Transformer models by adaptively clustering attention heads and FFNs, introducing the concept of learngenes.

Quality: The research is thorough, with a well-defined methodology and robust empirical validation across multiple datasets.

Clarity: The paper is clearly structured and articulated, making the complex methodology understandable.

Significance: It addresses an important problem in AI—efficient model scaling for different resource constraints, with potential far-reaching impacts in various application domains.

Creative Combination: The paper successfully combines existing concepts in a novel way, leading to a unique and effective model initialization approach.

Removing Limitations: It offers a solution that overcomes the limitations of traditional model initialization, enhancing flexibility and efficiency.

**Weaknesses:**

Lack of comparison to other clustering methods: The paper uses a density-based clustering approach, but does not compare this to other clustering methods like k-means.

Experiments focused on image classification: The experimental evaluation is primarily on image classification tasks. Testing on a broader range of vision tasks like object detection or segmentation would demonstrate the generalizability of the approach better.

The paper format can be improved: Table 2 should underline the highest result. Tables 4 and 5 should be three-line tables.

**Questions:**

1. Can you offer more experiments on other architectures(ConvNext, hybrid arch etc.), datasets(VTAB, FGVC etc.) or even modalities(like NLP tasks&models) to further validate the effectiveness of Cluster-Learngene? Which I believe will be powerful proof of its generalizability.

2. There have been other methods in model merging, can you compare Cluster-Learngene with these methods[1,2] to highlight its advantages, in experimental and theoritical ways?
[1]Yang, Enneng, et al. "Adamerging: Adaptive model merging for multi-task learning." arXiv preprint arXiv:2310.02575 (2023).
[2]Yang, Enneng, et al. "Representation Surgery for Multi-Task Model Merging." arXiv preprint arXiv:2402.02705 (2024).

3. How computationally expensive is the clustering step compared to the savings in downstream fine-tuning? Is there a break-even point in terms of number of downstream tasks where this becomes beneficial?

**Limitations:**

The authors provide a good discussion of limitations in the appendix, acknowledging that the performance of descendant models depends heavily on the quality of the ancestry model.

The discussion of broader impacts is quite limited. Given that the method aims to make transfer learning more efficient, some discussion of potential positive impacts (e.g. democratizing access to good models with less computational resources) as well as negative impacts (e.g. potential to amplify biases present in large pre-trained models) would be valuable. The authors could expand on mitigation strategies for some of these risks.

---

> ### Author Rebuttal · Authors · 2024-08-05
>
> Thank you for your thoughtful and constructive feedback. Below, I will respond to your questions.
>
> > Q1: Lack of comparison to other clustering methods.
>
> R1: Your question is thoughtful. In **Section 4.3.1**,  we have compared clustering methods such as k-means and provided different values for the set (k) of cluster centroids in k-means. The results are as follows:
>
> | *k-means*,  $k=1$ | *k-means*,  $k=2$ | *k-means*, $k=3$ | Ours      |
> | ----------------- | ----------------- | ---------------- | --------- |
> | 80.12             | 81.02             | 81.41            | **85.38** |
>
> Cluster-Learngene not only outperforms in clustering efficiency but also adaptively adjusts the count of cluster centroids for each model layer, unlike *k-means* which requires predefined numbers of cluster centroids.
>
>
>
>
>
>
>
> > Q2: Experiments focused on image classification.
>
> R2: Thanks for your suggestion. We conduct additional segmentation experiments on the ADE20K [A]. We set the base learning rate to $10^ {−3}$ and train for 160K iterations with a batch size of 8. The results are as follows:
>
> | Method                 | T-Params (M) | I-Params (M) | FLOP (G) | mIoU      |
> | ---------------------- | ------------ | ------------ | -------- | --------- |
> | Pretraining-Finetuning | 98.5         | 83.2         | 105      | 47.08     |
> | Heuristic-Learngene    | 57.8         | 42.5         | 105      | 40.12     |
> | Cluster-Learngene      | 74.7         | 59.4         | 105      | **48.30** |
>
> , where “T-Params" and “I-Params" correspond to the **T**otal number of parameters in the downstream/descendant models and the number of parameters **I**nherited into the downstream/descendant models, respectively. We validate the efficient cross-task initialization capability of our Cluster-Learngene. On the other hand,  Fine-tuning exhibits inferior performance due to the risk of negative transfer compared to our Cluster-Learngene. [A] Bolei Zhou, et al. "Semantic understanding of scenes through the ADE20K dataset." Int. J. Comput. Vis., 127(3):302–321, 2019.
>
>
> > Q3: The paper format can be improved.
>
> R3：We thank the reviewer for their constructive feedback and will enhance the paper by underlining the highest result in Table 2 and reformating Tables 4 and 5 into three-line tables for improved clarity and readability.
>
>
>
> > Q4: Can you offer more experiments?
>
> R4: Thanks for your suggestion.    (1) We have further supplemented the result for the **hybrid architecture**, such as LeViT-192 [B], which integrates convolutional and transformer components. We perform clustering and expansion on the MSA and FFN in stages 1-3 of LeViT-192. The experimental results are presented in the table below:
>
> | Method                 | CIFAR-100 | ImageNet  |
> | ---------------------- | --------- | --------- |
> | Pretraining-Finetuning | 85.11     | 69.08     |
> | From-Scratch           | 74.06     | 65.12     |
> | Heuristic-Learngene    | 78.22     | 66.65     |
> | Auto-Learngene         | 80.96     | 67.91     |
> | Cluster-Learngene      | **86.27** | **71.60** |
>
> We fine-tune all downstream models for 500 epochs on CIFAR-100 and 50 epochs on ImageNet. It can be observed that our method, when extended to the hybrid architecture, also achieves the best results on both CIFAR-100 and ImageNet.  [B] Graham, Benjamin, et al. "Levit: a vision transformer in convnet's clothing for faster inference." *ICCV*. 2021.
>
> （2）For Section 4.2.3, in addition to the existing results on FGVC (CUB-200-2011, Stanford Cars, Oxford Flowers) and VTAB (CIFAR-100, Flowers102)   datasets, we have also supplemented results from **FGVC** datasets such as Stanford Dogs, NABirds, and four **VTAB** datasets which are SVHN, DTD, EuroSAT, and Resisc45. The results are presented in the table below:
>
> | Method                 | Stanford Dogs | NABirds   | SVHN      | DTD       | EuroSAT   | Resisc45  |
> | ---------------------- | ------------- | --------- | --------- | --------- | --------- | --------- |
> | Pretraining-Finetuning | 75.41         | 78.33     | 97.57     | 84.88     | 97.32     | 96.53     |
> | From-Scratch           | 61.45         | 64.99     | 91.23     | 73.74     | 92.49     | 90.60     |
> | Heuristic-Learngene    | 71.68         | 73.12     | 94.81     | 77.30     | 94.88     | 94.25     |
> | Auto-Learngene         | 73.53         | 73.65     | 95.16     | 79.85     | 95.02     | 94.37     |
> | Cluster-Learngene      | **76.37**     | **80.76** | **97.92** | **85.30** | **98.24** | **96.85** |
>
> Our Cluster-Learngene outperforms Pretraining-Finetuning and other previous Learngene methods, thereby demonstrating the strong initialization capability and generalizability of our approach.
>
>
>
>
> > Q5: Can you compare Cluster-Learngene with model merging[1,2]?
>
> R5: Thank you to the reviewer for providing these papers. We will cite them in the 'Related Work' section. At the conceptual level, Multi-Task Model Merging[1,2] aims to merge models for collaborative multi-task processing, whereas our Learngene is designed to **independently initialize** diverse models and generate **scalable scales** to meet specific task demands and resource limitations. In the experimental  way, we compare our method with the two Multi-Task Model Merging methods[1,2] on the cars dataset. Our method achieves a performance of **89.87%** on DeiT-Small, outperforming the Multi-Task Model Merging methods[1,2], which show performances of 69.6% and 72.0% on the larger-scale ViT-B/32, respectively.
>
>
>
> > Q6: Computationally expensive and break-even point.
>
> R6:  （1）For the largest model, DeiT-Base, the computational time for the clustering step is **less than 6 minutes** on a single GeForce RTX 3090 GPU, which is negligible compared to the several hours or even a full day required for fine-tuning downstream tasks. （2）Indeed, our Cluster-Learngene reaches a break-even point with as few as two downstream tasks, indicating that the conditions for our algorithm to be beneficial are quite lenient.

---

> ### Comment · Reviewer_ScsP · 2024-08-09
>
> Thank you for addressing many of the concerns I initially had with your manuscript. However, I maintain my original score due to two persisting issues:
>
> Performance in Practical Settings: While the theoretical contributions of your work are clear, the practical performance does not appear to significantly surpass the traditional industry-standard approach of pre-training followed by fine-tuning. The advantages of your method over existing techniques are not clearly demonstrated, and the problem statement and solution approach described in the abstract seem somewhat superficial. For your work to have a substantial impact, it would be beneficial to provide more concrete evidence or comparative analysis demonstrating its superiority in real-world applications.
>
> Reproducibility Concerns: The complexity of your proposed method and the lack of critical implementation details raise concerns about reproducibility. Without these details, it may be challenging for readers to replicate your results or compare them with their work, which could undermine the credibility and utility of your paper. I would encourage you to include these essential details or simplify the approach to facilitate a better understanding and reproducibility by the wider research community.
>
> I hope these points are helpful for refining your paper.

---

> > ### Author Response · Authors · 2024-08-10
> >
> > Thank you very much for your timely response and constructive suggestions. Below, I will address the questions that you have newly raised:
> >
> > > Q1:Performance in Practical Settings
> >
> > We compared the traditional industry-standard approach of pre-training followed by fine-tuning and found that although our approach inherits fewer parameters, the performance is better. Therefore, this result needs to be viewed comprehensively and the representative results are shown in the table below:
> >
> > | Method                 | Params (M) | iNat-2019 | Cars      | **NABirds** |
> > | ---------------------- | ---------- | --------- | --------- | ----------- |
> > | Pretraining-Finetuning | 10.5       | 68.48     | 86.81     | 78.33       |
> > | Cluster-Learngene      | **7.5**    | **71.09** | **89.87** | **80.76**   |
> >
> >  In response to your request, we are pleased to report that in our exploration of real-world applications, we have extended our presentation beyond the classification results featured in the paper. Specifically, we have supplemented our experiments with segmentation results in the rebuttal phase, as per your guidance. Moreover, to initialize $N$ downstream models of different scales, Pretraining-Finetuning would require pre-training $N$ times. However, our method only needs to utilize **one** pre-trained model, from which we can cluster and expand to initialize $N$ downstream models of different scales, saving the pre-training time for $N-1$ models.
> >
> > > Q2: Reproducibility Concerns
> >
> > Due to the organizing committee's policy against providing anonymized code links directly in the rebuttal response, we have sent the anonymized code link to the Area Chair (AC) as per the instructions. We kindly ask the AC to forward it to you, in the hope that this will address your reproducibility concerns.

---

> > > ### Comment · Reviewer_ScsP · 2024-08-12
> > >
> > > While the method proposed has potential, I have concerns that I believe need to be addressed to meet the publication standards particularly regarding reproducibility and the practical significance of the work. Below, I outline my concerns and request further clarifications.
> > >
> > > ### Weaknesses:
> > >
> > > 1. **Reproducibility Concerns:**
> > >   My previous concern was not a request for code per se but rather an expression of apprehension regarding the reproducibility from the submission as presented. Even though the authors have now provided the code to the Area Chair, the submission in its current form does not meet the standards for acceptance based on reproducibility. I will lower my confidence rating and defer to the Area Chair to make a final judgment based on the quality of the provided code.
> > >
> > > 2. **Lack of Practical Significance:**
> > >    The practical significance of the proposed method is not sufficiently demonstrated. The manuscript fails to convincingly argue why the industry would adopt such a method in the face of increasingly popular large models like SAM and CLIP, which are becoming more efficient to fine-tune (e.g., through methods like LoRA and Adapters). The abstract does not adequately highlight the relevance and potential impact of the proposed method within this context.
> > >
> > > ### Specific Questions:
> > >
> > > - **Q1: Abstract and Generalization Concerns:**
> > >   The statement in the abstract, "However, the common practices often overgeneralize the applicability of these models, overlooking the task-specific resource constraints," is quite puzzling. Generalization is a cornerstone of success in deep learning. It appears there is a misunderstanding or miscommunication here, as models like CLIP are, in fact, increasingly being applied to specialized tasks with considerable success. Could you please clarify what is meant by overgeneralization in this context?
> > >
> > > - **Q2: Clarity and Coherence in Abstract:**
> > >   Frankly, the abstract is poorly written, with unclear problem statements and key solutions, which makes the entire paper somewhat perplexing. It is crucial for the abstract to clearly outline the challenges and articulate how the proposed method addresses these challenges. A revision to better communicate these aspects would significantly improve the paper's impact and understandability.
> > >
> > > **Closing Remarks:**
> > >
> > > I hope these comments are helpful for refining your paper. Enhancing these aspects could substantially improve the manuscript's clarity, practical relevance, and reproducibility, thereby strengthening its contribution to the field.

---

> > > > ### Author Response · Authors · 2024-08-12
> > > >
> > > > Thank you very much for your timely response and feedback. The specific questions you raised are excellent, as they facilitate our further clarification and help to dispel any misunderstandings. Below, I will respond to your questions.
> > > >
> > > > > Q1:Reproducibility Concerns.
> > > >
> > > > R1: We appreciate your focus on the reproducibility of our work and understand the importance of meeting the community's standards in this regard. We apologize for any confusion caused by our initial submission and are committed to addressing your concerns. To ensure reproducibility, we have provided the complete code to the Area Chair as per the guidelines. We have taken care to include all necessary details and dependencies to allow for the replication of our experiments and results.
> > > >
> > > >
> > > >
> > > > > Q2:Lack of Practical Significance.
> > > >
> > > > R2: Thank you for highlighting the practical significance of our proposed method. We recognize the importance of clearly presenting the unique advantages and potential impact of our approach. In our manuscript, we place a strong emphasis on the practical benefits of our method, which include:
> > > >
> > > > (1) **Scalability**: Our approach is designed to initialize downstream models of various scales to meet the needs of different downstream tasks. This level of scalability, which allows for a high degree of customization and adaptation to specific task requirements, is a characteristic that technologies such as LoRA and Adapters do not possess, as demonstrated through our results in Tables 1 and 2 and Figure 3.
> > > >
> > > > (2) **Resource Efficiency**: We underscore the resource optimization capabilities of our method, a critical feature for applications operating under constrained computational budgets, as demonstrated in Table 3 and Figure 5.
> > > >
> > > >
> > > >
> > > > > Q3: Abstract and Generalization Concerns
> > > >
> > > > R3: In the context of our work, by "overgeneralize ," we refer to the tendency to apply pre-trained models in a broad range of tasks without sufficient consideration for the unique requirements and constraints of specific applications. While models like CLIP have indeed shown remarkable versatility, our paper emphasizes the importance of customizing model initialization for downstream models of varying scales according to the particularities of each task, especially when there are resource constraints. We have designed experiments of different scales, as demonstrated in **Table 1** and **Figure 3** of the main paper, to showcase the effectiveness of our tailored initialization approach.
> > > >
> > > >
> > > >
> > > > > Q4:Clarity and Coherence in Abstract
> > > >
> > > > R4: We are grateful for your candid feedback on the clarity and coherence of our abstract. We have taken your comments to heart and will undertake a thorough revision of the abstract.  We will focus on defining the problem of task-specific resource constraints in the context of deep learning and succinctly describe how Cluster-Learngene offers a novel and effective strategy for model initialization that is both elastic in scale and sensitive to the needs of downstream tasks. In addition, we also hope that the explanations provided in R3 will aid in your understanding of our abstract and help clarify any misunderstandings.

---

> > > > > ### Comment · Reviewer_ScsP · 2024-08-14
> > > > >
> > > > > Thank you for your detailed rebuttal and the efforts to address the concerns raised. I appreciate the clarity provided in your responses, and it helps in better understanding the intentions and execution of your work. However, I would like to emphasize a couple of points to ensure that the manuscript aligns with the real-world applicability and theoretical positioning of the proposed method.
> > > > >
> > > > > Continuing Concerns:
> > > > > Understanding of Generalization:
> > > > > It appears there might be a misunderstanding regarding my previous comment on generalization. The success of widely adopted models like CLIP and SAM, which are finely tuned for various tasks including new categories in industrial applications, highlights a pivotal issue. Practically, most AI engineers would opt to fine-tune such established models rather than adopting a new method. The repeated emphasis on "generalization across different models" might be diverting from the practical effectiveness of widely used methods. A practical example, such as deploying a new industrial detection model, typically sees immediate application of methods like SAM. It may be beneficial to realign the discussion to how your method could complement or offer distinct advantages over these well-entrenched approaches in specific scenarios.
> > > > >
> > > > > Critique of Over-Generalization in Deep Learning:
> > > > > The critique of pretrained models' over-generalization in deep learning presented in your manuscript seems to create a problem that might not exist to the extent suggested. While academic research indeed isn't just about surpassing previous performance metrics, it's crucial to base arguments on clearly existing gaps or limitations in current methodologies. I believe it would strengthen your paper to reframe the narrative to emphasize the unique contributions of your method rather than positioning it against a seemingly exaggerated issue. This approach would likely resonate more authentically with the community and highlight the novel aspects of your work without undermining the established successes of deep learning generalization.
> > > > >
> > > > > Conclusion:
> > > > > Despite these concerns, I acknowledge the potential of the method and the effort to ensure reproducibility by providing code, albeit with some readability issues. Considering the comprehensive responses and revisions, I am inclined to adjust my score to a 5, reflecting my recognition of your method’s merits and my best wishes for your success in refining the paper to meet the community's standards and expectations.

---

> > > > > > ### Author Response · Authors · 2024-08-14
> > > > > >
> > > > > > Dear Reviewer,
> > > > > >
> > > > > >
> > > > > >
> > > > > > Thank you for the insightful comments and constructive suggestions you provided during the review process. Your expertise and attention to detail have significantly enhanced the quality of our manuscript. We are confident that our final submission will reflect the improvements discussed and will be better aligned with the interests and expectations of the community.
> > > > > >
> > > > > > Once again, thank you for your time, efforts, and for providing us with the opportunity to improve our work.
> > > > > >
> > > > > >
> > > > > >
> > > > > > Best regards,

---

### Official Review · Reviewer_SzCi · 2024-07-08

**Soundness:** 3
**Presentation:** 3
**Contribution:** 4
**Rating:** 8
**Confidence:** 5

**Summary:**

This paper seeks to address the issue of overgeneralizing the applicability of large pre-trained models in deep learning, particularly when faced with task-specific resource constraints. The authors propose Cluster-Learngene, a novel method that clusters critical internal modules from a large ancestry model and uses them to initialize descendant models of various scales. By adaptively clustering attention heads and feed-forward networks (FFNs) based on their density characteristics, Cluster-Learngene creates a flexible and efficient initialization process. The method also incorporates priority weight-sharing and learnable parameter transformations to expand the learngene. Extensive experiments demonstrate that Cluster-Learngene outperforms other initialization methods in efficiency and in customizing models to fit the resources available for downstream tasks.

**Strengths:**

1.	The problem studied in this paper is interesting and valuable. The paper is well-structured and clearly written, making it accessible and comprehensible to a broad audience.

2.	The approach of adaptively clustering attention heads and FFNs based on density characteristics is innovative and adds a novel dimension to model initialization techniques. The introduction of priority weight-sharing and learnable parameter transformations effectively addresses the need for models to adapt to varying resource constraints.

3.	The authors have conducted an extensive set of experiments to validate the effectiveness of Cluster-Learngene, including initializing descendant models of elastic scales and evaluating initialization results on different downstream tasks, both of which show significant improvements compared to baselines.

**Weaknesses:**

1.	The paper lacks a complexity analysis or specific time cost assessments for Cluster-Learngene, which are crucial for understanding its computational efficiency.

2.	Regarding the choice of hyperparameters, the authors could provide an ablation study, particularly for parameters like ε, to better understand their impact on the model's performance.

3.	Will the code be available for open source in the future?

**Questions:**

See Weaknesses.

**Limitations:**

The authors have addressed the limitations and potential negative societal impacts of their work.

---

> ### Author Rebuttal · Authors · 2024-08-05
>
> Thank you for your thoughtful and constructive feedback. Below, I will respond to your questions.
>
> > Q1: The paper lacks a complexity analysis or specific time cost assessments for Cluster-Learngene, which are crucial for understanding its computational efficiency.
>
> R1: For the largest model, DeiT-Base, the computational time for the clustering step is **less than 6 minutes** on a single GeForce RTX 3090 GPU, which is negligible compared to the several hours or even a full day required for fine-tuning downstream tasks.
>
>
>
> > Q2: Regarding the choice of hyperparameters, the authors could provide an ablation study, particularly for parameters like ε, to better understand their impact on the model's performance.
>
> R2: In the **Appendix A.6**, we have analyzed the varying settings of the hyperparameter Eps (ε). The following table displays the results for different values of ε:
>
> | $\varepsilon=1$ | $\varepsilon=10$ | $\varepsilon=100$ |
> | --------------- | ---------------- | ----------------- |
> | 85.27           | **85.38**        | 80.45             |
>
> $\varepsilon=1$ implies no FFN clustering, potentially causing negative transfer. $\varepsilon=100$ means clustering all FFNs in adjacent layers with identical head centroid counts, resulting in an excessive cluster of FFNs and subsequent degradation in the performance of initialized descendant models. Our Cluster-Learngene ($\varepsilon=10$) strikes a good balance between these issues.
>
>
>
> > Q3: Will the code be available for open source in the future?
>
> R3:  We expect to release our code by late Sep.

---

### Official Review · Reviewer_2s46 · 2024-07-12

**Soundness:** 2
**Presentation:** 2
**Contribution:** 2
**Rating:** 5
**Confidence:** 2

**Summary:**

The paper introduces Cluster-Learngene, a novel approach for initializing Vision Transformers (ViTs). This method clusters attention heads and position-wise feed-forward networks (FFNs) from a large "ancestry" model to form a condensed initialization core, termed "learngene." By leveraging the density characteristics of attention heads and FFNs, Cluster-Learngene effectively customizes models of elastic scales to meet the resource constraints of various downstream tasks. The approach not only preserves critical knowledge but also enhances efficiency by reducing redundancy and facilitating weight-sharing in descendant models.

**Strengths:**

- Originality: The clustering approach for attention heads based on density characteristics is innovative.
- Quality: The empirical validation is extensive, comparing the proposed method against several state-of-the-art initialization techniques.
- Clarity: Despite some dense sections, the overall exposition of concepts is clear.
- Significance: The method addresses critical issues in the scalability and resource efficiency of model deployment, making it highly significant for applications in resource-constrained environments.

**Weaknesses:**

Insufficient Downstream Task Evaluation: The dataset assessments for downstream tasks are currently limited to classification challenges. The generalizability of this method to other tasks such as segmentation and detection remains unknown.

**Questions:**

1. The Cluster-Learngene method is primarily designed for models with full attention mechanisms. Given the emergence of attention variants aimed at reducing computational costs, such as sparse or low-rank attention, it remains to be seen whether this method can be effectively transferred and applied to these types of models.
2. The hyperparameter Eps (ε) is currently set manually and plays a crucial role in the effectiveness of the clustering. Could the authors provide ablation study data to explore the robustness of the algorithm with varying settings of this parameter?

**Limitations:**

Please refer to weakness and questions

---

> ### Author Rebuttal · Authors · 2024-08-05
>
> Thank you for your thoughtful and constructive feedback. Below, I will respond to your questions.
>
>
>
> > Q1: Insufficient Downstream Task Evaluation
>
> R1: Thanks for your suggestion. We conduct additional segmentation experiments on the ADE20K [A]. We set the base learning rate to $10^ {−3}$ and train for 160K iterations with a batch size of 8. The results are as follows:
>
> | Method                 | T-Params (M) | I-Params (M) | FLOP (G) | mIoU      |
> | ---------------------- | ------------ | ------------ | -------- | --------- |
> | Pretraining-Finetuning | 98.5         | 83.2         | 105      | 47.08     |
> | Heuristic-Learngene    | 57.8         | 42.5         | 105      | 40.12     |
> | Cluster-Learngene      | 74.7         | 59.4         | 105      | **48.30** |
>
> , where “T-Params" and “I-Params" correspond to the **T**otal number of parameters in the downstream/descendant models and the number of parameters **I**nherited into the downstream/descendant models, respectively. We validate the efficient cross task initialization capability of our Cluster-Learngene. On the other hand,  Fine-tuning exhibits inferior performance due to the risk of negative transfer compared to our Cluster-Learngene. [A] Bolei Zhou, , et al. "Semantic understanding of scenes through the ADE20K dataset." Int. J. Comput. Vis., 127(3):302–321, 2019.
>
>
> > Q2: Given the emergence of attention variants aimed at reducing computational costs, such as sparse or low-rank attention, it remains to be seen whether this method can be effectively transferred and applied to these types of models.
>
> R2:  Thanks for your thoughtful question.  In general, sparse and low-rank attention mechanisms reduce the computational load through specific acceleration techniques. Sparse attention achieves this by limiting each query to compute attention with only a local set of keys. Meanwhile, low-rank attention decomposes the attention weight matrix into a low-rank form, such as $\mathbf{A} \approx \mathbf{P}^{\top} \mathbf{Q}$, where $\mathbf{P}$ and $\mathbf{Q}$ are low-rank matrices. However, they still produce complete attention head outputs. Therefore, our Cluster-Learngene can still calculate the Mean Attention Distance and apply it to these sparse or low-rank attention mechanisms.
>
>
>
> > Q3: The hyperparameter Eps (ε) is currently set manually and plays a crucial role in the effectiveness of the clustering.
>
> R3:   In the **Appendix A.6**, we have analyzed the varying settings of the hyperparameter Eps (ε). The following table displays the results for different values of ε:
>
> | $\varepsilon=1$ | $\varepsilon=10$ | $\varepsilon=100$ |
> | --------------- | ---------------- | ----------------- |
> | 85.27           | **85.38**        | 80.45             |
>
> $\varepsilon=1$ implies no FFN clustering, potentially causing negative transfer. $\varepsilon=100$ means clustering all FFNs in adjacent layers with identical head centroid counts, resulting in an excessive cluster of FFNs and subsequent degradation in the performance of initialized descendant models. Our Cluster-Learngene ($\varepsilon=10$) strikes a good balance between these issues.

---

> ### Comment · Area_Chair_Vgtr · 2024-08-13
>
> Dear Reviewer 2s46,
>
> This paper received mixed ratings. I would really appreciate it if you could check the authors' responses and post your further concerns (if there are still remaining concerns). Thank you so much!
>
> Your AC

---

### Official Review · Reviewer_GgZj · 2024-07-13

**Soundness:** 3
**Presentation:** 2
**Contribution:** 2
**Rating:** 4
**Confidence:** 3

**Summary:**

This paper introduces Cluster-Learngene, a weight initialization method for initializing downstream models of various sizes with pre-trained models. The proposed method is based on Learngene and includes the following two improvements: MSA and FFN centroid, which extract critical parameters and reduce redundancy; priority weight-sharing, which is able to initialize downstream models with varying number of attention heads.

**Strengths:**

(1)	The paper is overall well-written with sound illustrations;

(2)	As a weight initialization method, the proposed method makes sense and holds many potential applications;

(3)	Compared with Learngene, the proposed method can better adapt to downstream models of different scales.

**Weaknesses:**

(1)	Section 3.1 and L183-196 seems to have a messy formatting of formulas; for example, some letters are missing. This reduces the readability and overall quality of the paper;

(2)	Some key experiments were insufficient. For example, the counterpart of this work is the pre-existing Learngene methods, whereas in Table 1, only training from scratch is compared and in Table 2 results on ImageNet are also lacking;

(3)	Some details of the proposed method are not explained clearly enough (see Questions).

**Questions:**

(1)	After Adaptively Learngene Clustering, the number of layers obtained is variable. How to deal with that if it is not equal to the number of layers in the downstream model?

(2)	In priority weight-sharing (Figure 2), what if the number of head centroids is less than the number of head in the downstream model?

(3)	Are all parameters of the downstream model (inherited or not) updated (or fixed) during training?

(4)	In L222, “The learnable parameters in Eqns. (7) are implemented through a nonlinear mapping such as a neural network with the rectified linear units (ReLU)”, does this mean that $\hat{W}_t$ is more than just a parameter?

**Limitations:**

The limitations have been discussed.

---

> ### Author Rebuttal · Authors · 2024-08-04
>
> Thank you for your thoughtful and constructive feedback. Below, I will respond to your questions.
>
> > Q1: Section 3.1 and L183-196 seems to have a messy formatting of formulas;
>
> R1: We appreciate the reviewer's keen observation regarding Section 3.1 and lines 183-196. We apologize for any inconvenience caused and assure the reviewer that we will thoroughly revise this section to ensure all formulas are correctly formatted and legible. Below are the complete expressions of equations (1) and (2) in Section 3.1：
>
> ​							$\begin{align}\mathbf{A}^h=\text{Attention}(\mathbf{Q}_ h, \mathbf{K}_ h, \mathbf{V}_ h) = \text{softmax}\left(\frac{\mathbf{Q}_ {h}\mathbf{K}_ {h}^\top}{\sqrt{d_ k}}\right)\mathbf{V}_ {h}\end{align}$. 							    (1)
>
> ​							$\text{MultiHead}(\mathbf{Q}, \mathbf{K}, \mathbf{V}) = \text{Concat}(\mathbf{A}^1, \ldots, \mathbf{A}^H)\mathbf{W}^O$.	 								 (2)
>
> Here are the revisions made to  L183-196：
>
> - When $H_d$ is divisible by $c_l$: The weights of head centroids are shared $\frac{H_d}{c_l}$ times in sequence. For instance, centroids of weights $\mathbf{A}^{(L,1)}$ and $\mathbf{A}^{(L,2)}$ each share their weights across four attention heads, which are then directly assigned to eight attention heads of the descendant model in layer $L$.
>
> -  When $H_d$ is not divisible by $c_l$: The weights of the head centroids are sequentially shared $\lfloor \frac{H_d}{c_l}\rfloor$ times, followed by appending $\mathbf{A}^{(l,1)}, \ldots, \mathbf{A}^{(l,H_d \mod c_l)}$ at the end. As an illustration, we share the centroids of weights $\mathbf{A}^{(1,1)}, \ldots, \mathbf{A}^{(1,5)}$ once and then append $\mathbf{A}^{(1,1)}, \ldots, \mathbf{A}^{(1,3)}$, thus initializing eight attention heads of the descendant model in the first layer.
>
>
>
> For the attention heads in the descendant models, we introduce the hyperparameter $\omega = \frac{H_a}{H_d}$ to denote the factor by which the number of attention heads is reduced compared to the ancestry model. ...  According to the adjustments in the number of attention heads, the weights $\mathbf{W}^O$ of the projection layer are also proportionally pruned and then inherited by the descendant models.
>
>
>
> > Q2：Some key experiments were insufficient.
>
> R2: Thank you for the reviewer's suggestion. (1) We have added a strong comparative method called 'Pretraining-Finetuning' in Table 1, as shown in the table below：
>
> | Model | $H_d$ | $L_d$ | Params (M) | FLOPs (G) | From-Scratch | Pretraining-Finetuning | Ours      |
> | ----- | ----- | ----- | ---------- | --------- | ------------ | ---------------------- | --------- |
> | Tiny  | 3     | 12    | 5.7        | 1.2       | 61.44        | 66.36                  | **70.28** |
> | Small | 6     | 12    | 22         | 4.6       | 68.56        | 75.01                  | **78.43** |
> | Base  | 12    | 12    | 86.6       | 17.5      | 77.22        | 78.13                  | **79.50** |
>
> From-Scratch trains models for 100 epochs, while the Pretraining-Finetuning and our method only fine-tune the downstream models for 50 epochs.  Compared with the other two methods, our approach demonstrates the best initial performance across different scales.
>
> (2) In Table 2, we have supplemented the results on ImageNet as shown in the table below:
>
> | Method                 | I-Params | ImageNet  |
> | ---------------------- | -------- | --------- |
> | Pretraining-Finetuning | 10.5     | 68.85     |
> | From-Scratch           | 0        | 64.91     |
> | Heuristic-Learngene    | 5.6      | 66.44     |
> | Weight-Transformation  | 10.5     | 68.20     |
> | Auto-Learngene         | 10.5     | 67.78     |
> | Cluster-Learngene      | 7.5      | **71.36** |
>
> Our method continues to consistently outperform the baselines on the ImageNet dataset.
>
>
>
> > Q3: After Adaptively Learngene Clustering, the number of layers obtained is variable. How to deal with that if it is not equal to the number of layers in the downstream model?
>
> R3:  Your question is thoughtful. We have already addressed this issue in the **Appendix A.3**："According to the adjustments in the number of attention heads, the weights $\mathbf{W}^O$ of the projection layer are also proportionally pruned or expanded with the hyperparameter $\omega$ and then inherited by the descendant models."
>
>
>
> > Q4: In priority weight-sharing (Figure 2), what if the number of head centroids is less than the number of head in the downstream model?
>
> R4: Even if the number of head centroids is fewer than the number of heads in the downstream model, we can initialize the heads in the downstream model multiple times with priority weight-sharing of head centroids. Therefore, our priority weight-sharing strategy does not require any absolute size relationship between the number of head centroids and the number of heads in the downstream model.
>
>
>
> > Q5: Are all parameters of the downstream model (inherited or not) updated (or fixed) during training?
>
> R5: All parameters of the downstream model (inherited or not) update during training. However, due to leveraging  weight-sharing, the computational cost of our descendant models remains smaller compared to fine-tuning the entire pre-trained model.
>
>
>
> > Q6: In L222, “The learnable parameters in Eqns. (7) are implemented through a nonlinear mapping such as a neural network with the rectified linear units (ReLU)”, does this mean that 𝑊^𝑡 is more than just a parameter?
>
> R6: I apologize for any confusion caused by the description in my paper.  The symbol  $\mathbf{\widehat{W}}_ {t}$ indeed represents the newly introduced learnable parameters that are used to expand the $t^{th}$ feedforward network (FFN). However, in the implementation of this expanding FFN process, not only do we apply the simple transformation with these learnable parameters  $\mathbf{\widehat{W}}_ {t}$, but we also incorporate the ReLU (Rectified Linear Unit) activation function to provide some additional non-linear transformations.

---

> ### Comment · Area_Chair_Vgtr · 2024-08-13
>
> Dear Reviewer GgZj,
>
> This paper received mixed ratings. I would really appreciate it if you could check the authors' responses and post your further concerns (if there are still remaining concerns). Thank you so much!
>
> Your AC

---

### Author Rebuttal · Authors · 2024-08-06

> Q1: Experiments focused on image classification: The experimental evaluation is primarily on image classification tasks. Testing on a broader range of vision tasks like object detection or segmentation would demonstrate the generalizability of the approach better.

R1: Thanks for your suggestion. We conduct additional segmentation experiments on the ADE20K [A]. We set the base learning rate to $10^ {−3}$ and train for 160K iterations with a batch size of 8. The results are as follows:

| Method                 | T-Params (M) | I-Params (M) | FLOP (G) | mIoU      |
| ---------------------- | ------------ | ------------ | -------- | --------- |
| Pretraining-Finetuning | 98.5         | 83.2         | 105      | 47.08     |
| Heuristic-Learngene    | 57.8         | 42.5         | 105      | 40.12     |
| Cluster-Learngene      | 74.7         | 59.4         | 105      | **48.30** |

, where “T-Params" and “I-Params" correspond to the **T**otal number of parameters in the downstream/descendant models and the number of parameters **I**nherited into the downstream/descendant models, respectively. We validate the efficient cross task initialization capability of our Cluster-Learngene. On the other hand,  Fine-tuning exhibits inferior performance due to the risk of negative transfer compared to our Cluster-Learngene. [A] Bolei Zhou, , et al. "Semantic understanding of scenes through the ADE20K dataset." Int. J. Comput. Vis., 127(3):302–321, 2019.



> Q2: The hyperparameter Eps (ε) is currently set manually and plays a crucial role in the effectiveness of the clustering.

R2:  In the **Appendix A.6**, we have analyzed the varying settings of the hyperparameter Eps (ε). The following table displays the results for different values of ε:

| $\varepsilon=1$ | $\varepsilon=10$ | $\varepsilon=100$ |
| --------------- | ---------------- | ----------------- |
| 85.27           | **85.38**        | 80.45             |

$\varepsilon=1$ implies no FFN clustering, potentially causing negative transfer. $\varepsilon=100$ means clustering all FFNs in adjacent layers with identical head centroid counts, resulting in an excessive cluster of FFNs and subsequent degradation in the performance of initialized descendant models. Our Cluster-Learngene ($\varepsilon=10$) strikes a good balance between these issues.



> Q3: How computationally expensive is the clustering step compared to the savings in downstream fine-tuning? Is there a break-even point in terms of number of downstream tasks where this becomes beneficial?

R3:  （1）For the largest model, DeiT-Base, the computational time for the clustering step is **less than 6 minutes** on a single GeForce RTX 3090 GPU, which is negligible compared to the several hours or even a full day required for fine-tuning downstream tasks.（2）Indeed, our Cluster-Learngene reaches a break-even point with as few as two downstream tasks, indicating that the conditions for our algorithm to be beneficial are quite lenient. Considering that real-world applications always involve multiple tasks, our approach provides clear benefits.



> Q4: The discussion of broader impacts.

R4: In practice, learngene strives to preserve the core knowledge from the original model, avoiding the redundancy found in large pre-trained models. Consequently, descendant models initialized with the learngene reduce the biases that are present in the original large pre-trained models.

---

### Decision · Program_Chairs · 2024-09-25

**Decision:**

Accept (poster)

**Comment:**

In this paper, the authors introduce Cluster-Learngene, a novel method for initializing vision transformers. After the discussion period, most of the reviewers tend to accept this paper. Reviewer GgZj still kept their negative rating. However, they didn't respond to the authors' feedback. After checking the paper, the reviews, and the discussions, I tend to accept this paper. The authors are required to include the discussions with the reviewers in their final version. Besides, the code shared during the discussion period should also be made publicly available.